# Amalgam plays a dual role in controlling the number of leg muscle progenitors and regulating their interactions with the developing *Drosophila* tendon

**Blandine Moucaud, Elodie Prince, Elia Ragot, Yoan Renaud, Krzysztof Jagla, Guillaume Junion, Cedric Soler** *

GReD Institute, UMR CNRS 6293, INSERM U1103, University of Clermont-Auvergne, Clermont-Ferrand, France

* cedric.soler@uca.fr

**Data Availability Statement:** All relevant data are within the paper and its Supporting Information

## Abstract

Formation of functional organs requires cell–cell communication between different cell lineages and failure in this communication can result in severe developmental defects. Hundreds of possible interacting pairs of proteins are known, but identifying the interacting partners that ensure a specific interaction between 2 given cell types remains challenging. Here, we use the *Drosophila* leg model and our cell type-specific transcriptomic data sets to uncover the molecular mediators of cell–cell communication between tendon and muscle precursors. Through the analysis of gene expression signatures of appendicular muscle and tendon precursor cells, we identify 2 candidates for early interactions between these 2 cell populations: *Amalgam (Ama)* encoding a secreted protein and *Neurotactin (Nrt)* known to encode a membrane-bound protein. Developmental expression and function analyses reveal that: (i) Ama is expressed in the leg myoblasts, whereas Nrt is expressed in adjacent tendon precursors; and (ii) in Ama and Nrt mutants, myoblast-tendon cell–cell association is lost, leading to tendon developmental defects. Furthermore, we demonstrate that Ama acts downstream of the FGFR pathway to maintain the myoblast population by promoting cell survival and proliferation in an Nrt-independent manner. Together, our data pinpoint Ama and Nrt as molecular actors ensuring early reciprocal communication between leg muscle and tendon precursors, a prerequisite for the coordinated development of the appendicular musculoskeletal system.

## Introduction

Development of organs in a multicellular organism requires the orchestration of cell–cell interactions that control the assembly of various cell types. Altered cellular communication or inappropriate responses to intercellular signals can lead to major developmental defects [1]. Research on multicellular organism development generally focuses on one specific tissue.

files. Except RNAseq metadata that are available from GEO database (accession number GSE245192).

**Funding:** This work was supported by AFM-Téléthon (MyoNeurAlp Strategic Program to CS), the Agence Nationale pour la Recherche (ANR LIMBCT to KJ) and the iSITE (CAP2025 Grant to CS). The funders had no role in study design, data collection and analysis, decision to publish, or preparation of the manuscript.

**Competing interests:** The authors have declared that no competing interests exist.

**Abbreviations:** APF, after pupae formation; CT, connective tissue; DE, differentially expressed; FC, fold change; GO, gene ontology; MCT, muscle connective tissue; RTK, receptor tyrosine kinase.

Classically, these studies highlight how the activation of signaling pathways, in a cell-autonomous or non cell-autonomous manner, influences the fate or behavior of a particular cell type. However, bi-directional communication between different cell types, and how these cells are integrated into a functional physiological unit, have been less investigated.

The limb musculoskeletal system is a remarkable model to study the integration of multiple cell types. It is composed of muscle fibers, motor neurons, blood vessels, and connective tissue (CT) cells. CTs include tendons connecting muscles to bones and irregular muscle CT (MCT) cells surrounding and connecting muscle fibers. During vertebrate development, CT progenitors from the lateral plate mesoderm and myogenic cells from the myotome undergo a coordinated migration into the limb bud and differentiate coordinately to build the appendicular musculoskeletal organ. Flies do not possess an internal skeleton. Instead, their muscles are connected to the exoskeleton through specialized muscle attachment cells known as apodemes. These structures are regarded as the functional counterparts of tendons found in vertebrates. Tendon and muscle progenitors of the *Drosophila* larval muscle system develop independently from each other [2]. However, in the *Drosophila* leg, the behavior of tendon and muscle progenitors are intimately linked [3,4]. This suggests that, like in vertebrates, there are reciprocal interactions between muscle and CT progenitors shaping the appendage musculature [5–7].

*Drosophila* leg musculature develops from the appendicular muscle precursors that lie on the surface of the leg disc in close vicinity of the leg tendon precursors that derive from the disc epithelium. At the onset of metamorphosis, tendon precursors undergo a collective cell migration and form long internal structures inside the developing leg, whereas myoblasts aggregate and engage in a coordinated migration following the invaginating tendon precursors. Consequently, the perturbation of tendon formation during the early steps of leg development affects the spatial localization of the associated myoblasts [7]. Interestingly, *Drosophila* leg myoblasts express *ladybird (lb)*, the vertebrate ortholog of *Lbx1*, which is also expressed in vertebrate myoblasts and required for their migration into the limb buds [8–10]. Moreover, appendicular tendon specification and invagination/migration rely on Stripe (Sr), Dar1 and Odd-skipped (Odd), 3 transcription factors whose vertebrate orthologs are markers of limb tendon and MCT cells [11,12]. Strikingly, in the limb of chick embryo, a subset of MCT cells expressing Odd skipped-related 1 (Osr1), the ortholog of Odd, were observed in close association with migrating Lbx1+ myogenic progenitors [11]. In *Osr1* mutant, the pool and distribution of myogenic cell is altered, suggesting that muscle progenitors require Osr1+ MCT cells to survive/proliferate and/or to reach the correct position [11].

Thus, convergent observations in vertebrates and invertebrates suggest the existence of a common genetic circuitry that controls muscle and CT progenitor interactions, and coordination during appendicular development. The molecular signals that ensure the specificity of interactions between subpopulations of myoblasts and their respective tendon precursors remain largely unknown. This prompted us to use the *Drosophila* model to search for candidates affecting early interactions between muscle and tendon precursors.

Nrt is a transmembrane protein of the serine esterase-like family and belongs to the family of neuronal cell adhesion molecules [13–17]. Ama is a secreted protein of the immunoglobulin superfamily [18] that plays a role as a cell adhesion molecule involved in axon guidance [15,16] by acting through its ligand, Nrt. In the central nervous system, Ama/Nrt also regulate neuroblast specification [19], and Ama affects glial cell proliferation/survival and migration through the receptor tyrosine kinase (RTK) signaling by modulating the expression of Sprouty, a negative regulator of the RTK pathway [20–22]. More recently, 2 independent single-cell transcriptomic studies indicated that Ama and Nrt are expressed in a subset of flight muscle progenitors. In the wing disc, depletion of Ama results in a reduced pool of wing disc-associated myoblasts [23], and Nrt is critical for proper direct flight muscle development [24].

Here, we generate transcriptomic data sets from myogenic precursors and analyze them together with our previously published data sets on isolated tendon precursors [25] and identify Neurotactin (Nrt) and its binding partner, Amalgam (Ama), as candidates selectively expressed in a specific tendon cluster and in leg myoblasts, respectively. We show that knockdown of Ama results in a dramatic decrease of the leg disc myoblast population independently of Nrt. However, later during metamorphosis, Ama and Nrt are both required for keeping tendon and myogenic precursor cells closely associated, thus ensuring correct development of appendicular muscles. These results highlight an unexpected double role of Ama during leg muscle formation in both maintaining the pool of leg muscle precursor cells and coordinating tendon-muscle precursor growth.

## Results and discussion

### Transcriptomics data identify Ama and Nrt as a candidate pair for cell–cell interaction

We previously reported transcriptional signatures of leg tendon precursor cells [25] at the developmental time point of 0 h after pupae formation (APF) when subpopulations of myoblasts localize around clusters of tendon precursors. Here, we analyzed the transcriptome of leg disc myoblasts at the same time of development to identify potential interacting pairs. After fluorescence-activated cell sorting and RNA sequencing of the myoblasts (see Material and methods and GSE245192), we performed differential gene expression analysis and identified a set of 2,236 differentially expressed (DE) genes in myoblasts compared to all other cells in the leg disc (Fig 1A and S1 Data). Among enriched genes (FC > 1.5), we consistently found genes specifically expressed in leg myoblast or involved in leg muscle development, such as *twist*, *htl*, *zfh1*, *him*, *mef2*, *vg*, *cut* [3,10,26–30]. Gene ontology (GO) analysis of enriched genes showed an overrepresentation of GO terms related to muscle development (Fig 1B). We then searched for corresponding interacting partners expressed in muscle and tendon precursors. Considering an RPKM>5, we identified up to 290 pairs of cell surface or secreted proteins known to interact with each other. To identify pairs of interactors more likely to participate in specific interactions between tendon cells and myoblasts, we narrowed down this list to genes exhibiting a more specific expression (FC > 1.5) in tendon cells and myoblasts compared to other disc cells (Fig 1C). Among the most enriched genes, we found *Nrt* encoding a transmembrane protein (RPKM = 234, FC = 2.5) specifically in tendon cells and *Ama* encoding its binding partner. *Ama* was highly enriched both in myoblasts (RPKM = 385, FC = 3) and in tendon cells (RPKM = 221, FC = 1.7). These data prompted us to analyze in detail the expression pattern of *Ama* and *Nrt* during leg disc development.

### Ama and Nrt show a complementary expression pattern in muscle and tendon precursors of the leg disc

As tendon precursors in the leg disc are specified from third larval instar (L3) and elongate during early metamorphosis, we performed immunostaining using anti-Nrt antibody on leg discs from L3 to 5 h APF expressing Lifeact.GFP in developing tendons (Sr>Lifeact.GFP line; Laurichesse and colleagues) (Fig 2A–2D and 2I–2L). Remarkably, in L3 stage, *Nrt* is expressed in only one cluster of tendon cells, in the dorsal femur, corresponding to the future tibia levator tendon (*tilt*) (for nomenclature, see Soler and colleagues) (Fig 2A and 2B). This tendon precursor-associated Nrt expression remains restricted to the dorsal femur in the next stages of development (Fig 2C, 2D, 2I, and 2J). In addition, Nrt is detected in the chordotonal organ (Fig 2A–2D, 2I, and 2J) and in the 2 most distal neuronal precursors that project axons in tarsi

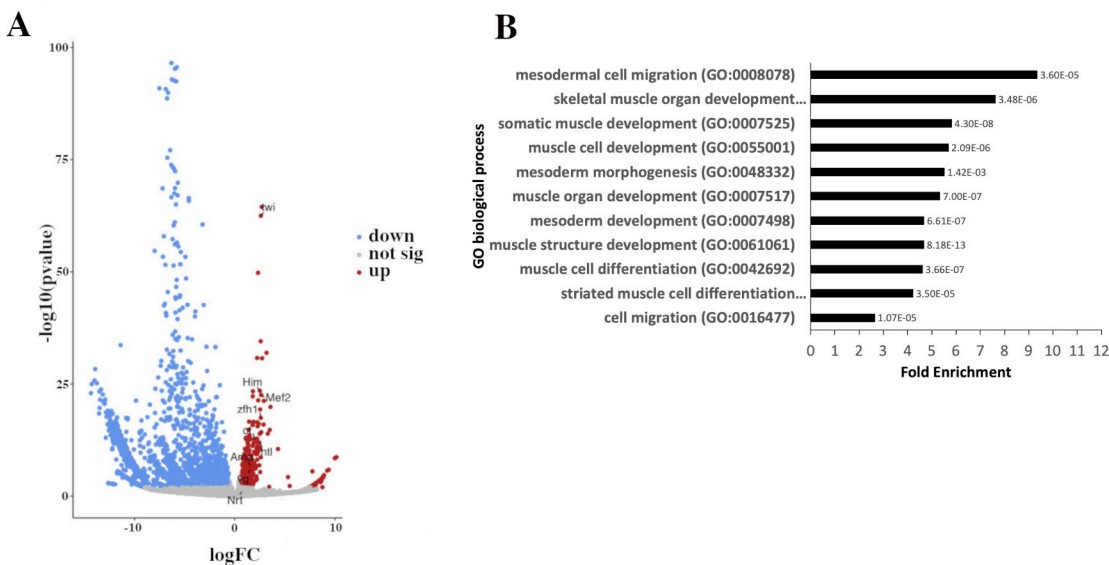

**Fig 1. DE genes and interactome data in leg disc myoblasts.** (A) Volcano plot showing DE genes, including 533 enriched genes (red) and 1,703 down-regulated genes (blue). Those not significantly changed (fold change < 1.5; $P > 0.05$) are in gray (S1 Data). Examples of genes known to be specifically expressed in myoblasts are annotated, as are *Nrt* and *Ama*. For graphical representation, outliers (-log10 (*p*-value) >100 and log FC >20) were excluded. Generated using Volcano Plot tool (Galaxy V.0.0.5). **(B)** GO analysis of enriched genes in leg myoblasts. Graph shows selected GO terms for biological process, GO term accession numbers, fold enrichment, and *p*-value. Data were obtained by GO overrepresentation test using PANTHER. **(C)** List of identified pairs of interacting partners expressed in myoblasts and tendon cells (Flybase protein interactions browser). Among the 290 pairs of interacting genes expressed in myoblasts and tendon cells (RPKM>5), 9 of them have corresponding interacting genes with a Fold Change >1.5. DE, differentially expressed; GO, gene ontology.

[31]. To follow *Ama* expression, we generated an *Ama*::*EGFP* knock-in line. In stages preceding tendon cell specification, Ama::EGFP protein can already be detected in the vicinity of leg disc myoblasts (S1A–S1F Fig). From L3 stage to 5 h APF, Ama is associated with muscle precursors located in the presumptive femur, tibia, and tarsi segments (Fig 2E–2H, 2K, and 2L), and is barely detectable around myoblasts in more proximal regions, at the periphery of the leg disc. This observation suggests that distal myoblasts could be the major source of secreted Ama protein. Noticeably, as early as L3 stage, Ama protein strongly accumulates in the dorsal femur at the interface with Nrt-expressing tilt tendon (arrowheads in Fig 2E–2H, 2K, and 2L). To accurately document the expression of *Ama* and *Nrt* in this specific region, we used confocal Airyscan technology to image at high spatial resolution, 5 h APF leg discs from a Sr>CAAXmCherry; *Ama*::*EGFP* line immunostained for the myoblast (Twist) and tendon (mCherry) markers and for Ama (EGFP) and Nrt (Fig 2M and 2N and 2M' and 2N'). High-

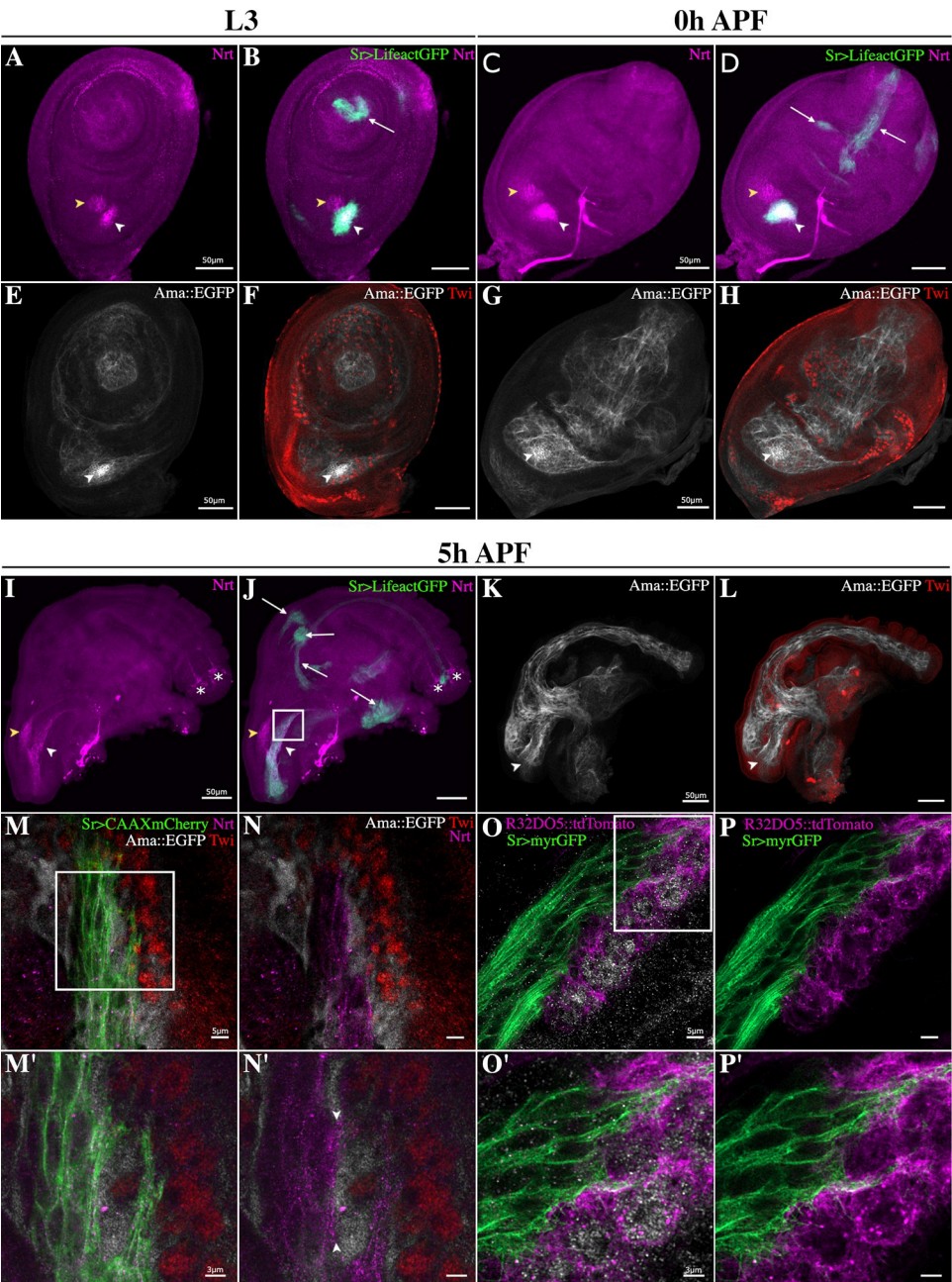

**Fig 2.** *Nrt* and *Ama* are expressed in muscle and tendon precursors during leg disc development. (A–L) Confocal optical sections of Sr-Gal4>UAS-Lifeact.GFP (green) and Ama::EGFP (gray) leg discs at different stages of development immunostained for Nrt (magenta) and for Twist (Twi—red), respectively. (A–D) In L3 and 0 h APF leg discs, Nrt is localized in the tilt tendon in the dorsal femur (white arrowhead) but not in the other tendon precursors in green (white arrows); note Nrt presence in the chordotonal organ (yellow arrow head). (E–H) In L3 and 0 h APF leg discs, Ama::EGFP protein is found surrounding most of distal myoblasts immunostained for Twi (red); but not around the most proximal ones at the periphery of leg disc. Ama::EGFP accumulates in the region of dorsal femur close to tilt tendon precursors (white arrowhead). (I, J) At 5 h APF, Nrt remains localized in the tilt that elongates in the dorsal femur (white arrowhead) and in the chordotonal organ (yellow arrowhead). Nrt is not visible in any tendons of the other segments (white arrows) but can be detected in the 2 most distal neuronal precursors projecting axons in tarsi (asterisks, Jan and colleagues). (K, L) At 5 h APF, Ama::EGFP is detected in the entire leg disc cavity where myoblasts stand, including the dorsal femur where the Nrt-expressing tilt tendon invaginates and elongates (arrowhead). (M–P') High-resolution imaging of Nrt-expressing tilt tendon and surrounding myoblasts at 5 h APF using Zeiss Airyscan confocal technology. The imaged regions on these discs approximately correspond to the framed area of the disc shown in J. (M, N) Close up view of myoblast-tendon interface of a 5 h APF leg disc expressing *Sr-*

*Gal4>CAAXmCherry* and *Ama*::EGFP immunostained with Twi and Nrt antibodies with higher magnifications (**M',** **N'**) of the framed area in **M**. Ama::EGFP (gray) localizes all around myoblasts (red) and next to the membrane of tendon cells (green), whereas Nrt (magenta) is specifically found in tendon cells. Note the enrichment of Nrt at the tendon membrane (between arrowheads in **N'**) facing the myoblast where Ama::EGFP accumulates. (**O, P**) Close up view of myoblast-tendon interface of a 5 h APF leg disc expressing *Sr-Gal4>myrGFP* and *R32D05::tdTomato* immunostained with Twi antibody with higher magnifications (**O', P'**) of the framed area in **O**. Labeled membranes of myoblasts (magenta) and tendon cells (green) show a closed proximity of these 2 cell types with cytoplasmic projections from tendon cells navigating between myoblasts. Note the relative short distance between the plasma membrane and the immunostaining of Twist transcription factor suggesting that in these cells nuclei occupy most of the cell volume. APF, after pupae formation.

resolution imaging of dorsal femur region reveals close association of myoblasts with the invaginating tendon (Fig 2M and 2N). A closer view confirms that Nrt protein is specifically expressed by the developing tendon and concentrate at the membrane of cytoplasmic projections facing myoblasts (Fig 2M' and 2N'). In contrast, Ama::EGFP staining accumulates at the interface between the myoblasts and the membrane of the tilt tendon cells exhibiting a more diffuse pattern, as expected for a secreted protein (Fig 2M' and 2N'). This corroborates our previous findings that myoblasts align along this tendon, and disruption in tendon development impact their spatial distribution. To further investigate the membrane proximity of these 2 cell types, we identified a new myoblast-specific Gal4 line, R32D05-Gal4 (S1G–S1I Fig; [32]) to generate a transgenic line expressing a membrane tagged tdTomato fluorescent protein under the R32D05 regulatory region (R32D05::mCD4tdTomato). This line was crossed with the Sr-Gal4 line expressing a membrane tagged myrGFP protein (Sr>UAS-myrGFP), and 5 h APF leg discs from the F1 generation were immunostained using anti-Twist antibody. High-resolution images, using Airyscan technology, confirmed the close proximity of the 2 membranes, highlighting cytoplasmic filopodia from tendon cells weaving between myoblasts (Fig 2O and 2P and 2O' and 2P'). These observations support the hypothesis of physical interaction between the myoblasts and developing tendons.

To precisely determine the origin of *Ama* expression, we performed a series of in situ hybridizations to detect *Ama* RNA in leg discs (Fig 3). We used the myoblast-specific Gal4 line R32D05-Gal4 crossed with an UAS-GFP line. In L3 larval leg disc of R32D05>GFP line, *Ama* RNA colocalizes with most GFP+ myoblasts as well as in 3 distinct clusters in tarsal segments, revealing an additional source of Ama in most distal segments (Fig 3A–3C). In the dorsal femur, in the vicinity of Nrt+ tendon, myoblasts show a particularly high level of *Ama* RNA. To confirm that *Ama* expression is myoblast-specific, we crossed a UAS-AmaRNAi line with R32D05-Gal4>UAS-GFP line [32]. Ama RNAi expression in myoblasts led to a strong reduction of *Ama* RNA level in leg disc, except in the 3 clusters in the tarsi where R32D05-Gal4 driver is not expressed (Fig 3D–3F). In the dorsal femur, *Ama* expression is completely abolished indicating that at L3 stage, myoblasts are the only source of Ama in this region. However, at later stage of development (5 h APF), *Ama* RNA was clearly detectable in both myoblasts and in the tilt tendon in control leg disc (S2A–S2C and S2G–S2I Fig). Accordingly, when we used specific Gal4 drivers to induce UAS-AmaRNAi either in myoblasts (R32D05-Gal4) or tendon (Sr-Gal4), *Ama* expression was abolished only in myoblasts or only in the tendon (S2D–S2F and S2J–S2L Fig). This result indicates that, a few hours after pupae formation, the tendon in the dorsal femur also expresses *Ama*. To determine whether myoblasts are the sole source of the Ama protein in the dorsal femur, we examined the presence of the Ama protein in R32D05-Gal4>UAS-AmaRNAi discs, where most of the myoblasts are missing at 5 h APF. To mitigate any potential effects of the GFP peptide on the stability of the Ama::EGFP protein, we used an antibody that directly targets the Ama protein [33]. When *Ama* expression is specifically reduced in myoblasts, the Ama protein level is barely detected in the femur region at 5

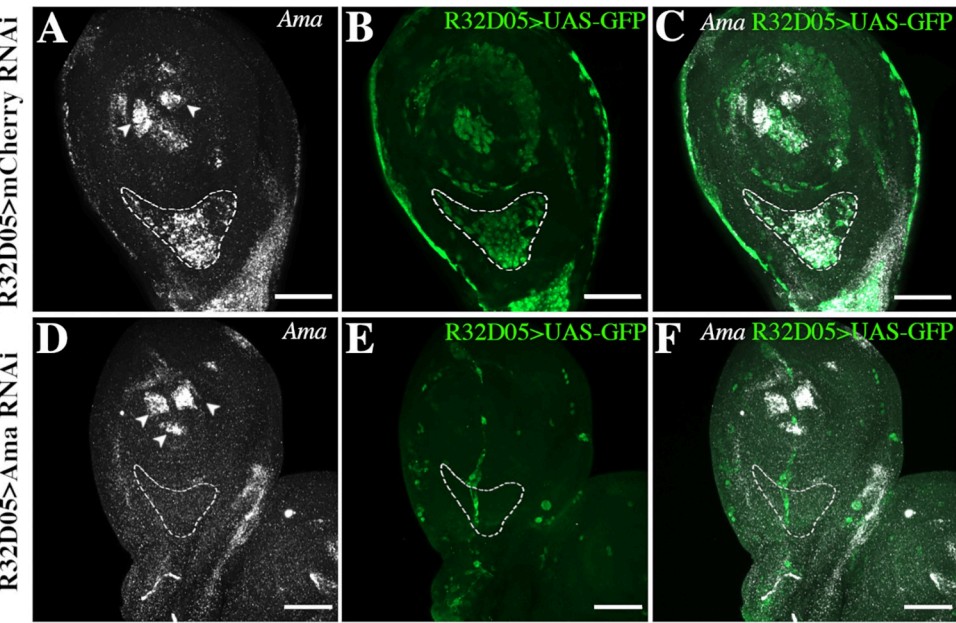

**Fig 3. Myoblasts are a main source of Ama in third larval instar.** Expression of *Ama* revealed by in situ hybridizations on L3 leg discs from R32D05-Gal4>UAS-GFP larvae. **(A–C)** In control leg disc expressing *UAS-mCherryRNAi* in the myoblasts, *Ama* mRNA localized in myoblasts (green) with a sustained expression in the dorsal femur (outlined area). Note the 2 additional sources of *Ama* mRNA in the most distal tarsus that are not GFP-positive (arrowheads in A). **(D–F)** In leg disc expressing *UAS-AmaRNAi* specifically in the myoblasts, *Ama* expression is nearly abolished in the region of the tilt in the dorsal femur (outlined area); confirming that, in this region, myoblasts are the main source of Ama transcripts at this stage. As an internal control, *Ama* remains strongly expressed in the 3 non-GFP clusters of the distal tarsus (arrowheads). Scale bar 50 μm.

h APF (**S2M–S2P Fig**). This suggests that, although *Ama* transcripts are detected in tilt at 5 h APF, the myoblasts remain the major source of Ama protein at this stage.

Overall, our expression analyses indicate that most myoblasts express *Ama*, while *Nrt* is restricted to a single developing tendon (tilt). *Ama* expression is also initiated earlier in larval development than that of Nrt and could be detected in leg myoblasts at L2 larval stage (**S1A–S1F Fig**). Thus, spatial and temporal expression patterns of Ama and Nrt suggest that during appendicular myogenesis Ama could act not only by interacting with Nrt but also in a Nrt-independent way.

## Ama controls myoblast number in an Nrt-independent way

Ama was previously shown to regulate the pool of glial cells and flight muscle progenitors [22,23]. Interestingly, Ama is expressed in leg disc myoblasts from the early larval stage onwards (S1A–S1C Fig). Ama down-regulation in myoblasts in R32D05-Gal4>UAS-GFP leg discs (Fig 3) leads to a strong reduction in GFP fluorescence, suggesting that the number of myoblasts could have been reduced. Because the number of total leg disc myoblasts and their proliferative rate have never been precisely determined, we first counted them at different stages of development using a Twist antibody. The number of myoblasts grows from a pool of around 10 per disc primordia at the end of embryonic stage to 36 (+/−10) at mid-L2 stage and to 653(+/−82) by the end of L3 stage.

To thoroughly assess the effect of Ama knockdown on the pool of myoblasts, we used a Twist antibody to compare the number of muscle progenitors in L3 leg discs from R32D05-Gal4>UAS-AmaRNAi, R32D05-Gal4>UAS-mcherryRNAi, and w[1118] strains.

Myoblast-specific RNAi-knockdown of *Ama* leads to a severe reduction in muscle precursors with almost no Twist+ myoblasts (<10/disc) when compared to both controls (Fig 4A, 4B, and 4E and S2 Data). No significant differences could be observed between R32D05-Gal4>UAS-mcherryRNAi and w[1118] controls, with 643 and 653 myoblasts on average, respectively, by the end of L3. Thus, Ama depletion results in a severe reduction of Twist+ myoblasts associated with the leg disc. As a decrease in the number of cells may be due to a defect in cell proliferation and/or to an increase in cell death, we tested these 2 variables by immunostaining L3 leg discs with antibodies against either the mitotic marker phospho-histone H3 (pH3) or the apoptotic marker caspase-activated Dcp1. Because these 2 antibodies were raised in rabbit, as was the Twist antibody, we could not use this latter to visualize the myoblasts. So, we performed the anti-Dcp1 and anti-pH3 immunostainings on R32D05-Gal4>UAS-AmaRNAi and R32D05-Gal4>UAS-mcherryRNAi leg discs expressing UAS-GFP to visualize the myoblasts (Fig 4F–4J). Then, we quantified the number of GFP cells that were pH3+ or Dcp1+. In control leg discs, 1.8% of cells were pH3+/GFP+. After *Ama* RNAi induction, this ratio decreased at 0.7% (Fig 4F–4H and S2 Data), and 2.5% of GFP+ cells were also Dcp1+ in controls, indicating the cells were apoptotic. This ratio increased to 13.8% when *Ama* was knocked down (Fig 4H–4J). The low number of total myoblasts in L3 leg discs after Ama knockdown (<10) can make these outcomes difficult to interpret. Thus, to increase the number of myoblasts at the time of counting (L3), we delayed the expression of the *UAS-AmaRNAi* transgene using Gal80[ts], a thermosensitive form of the Gal4-inhibitor Gal80 [34]. In this way, we induced *UAS-AmaRNAi* expression from L2 stage, and we could count an average of 100 GFP+ cells in L3 stage. The percentage of mitotic cells was still significantly lower compared to controls, and the percentage of apoptotic cells was higher (Fig 4H), indicating that Ama depletion affects both viability and the proliferation rate of leg disc myoblasts. Since *Ama* expression was also observed in tendon cell (Figs 2 and S2), we tested whether AmaKD could similarly impact the viability and the proliferation rate of these cells. However, anti-Ph3 and anti-Dcp1 immunostainings showed no difference between Sr-Gal4>UAS-AmaRNAi and control leg discs (S3A–S3C Fig).

A previous study suggested that Ama-depleted flight muscle precursors were unable to proliferate but the authors could not assay a defect in proliferation rate [23]. Moreover, Nrt was expressed in flight muscle precursors and its depletion resulted in a flight muscle defect, suggesting that Ama may interact with Nrt to regulate the pool of flight muscle myoblasts [23,24]. To test this possibility in leg myoblasts, we counted the number of Twist+ cells in leg discs of Nrt[1]/Nrt[2] transheterozygous mutants. We observed no differences compared to w[1118] or with heterozygous Nrt[1]/Nrt[+] and Nrt[2]/Nrt[+] leg discs (Fig 4C–4E). This result is consistent with the absence of *Nrt* expression in myoblasts at L3 stage (Fig 2). Moreover, myoblast depletion can already be observed in Ama-RNAi expressing leg discs of earlier stage L2 (S3A–S3F Fig), a stage when Nrt could not be detected in leg disc even in tendon cells, which are specified at the beginning of L3 stage (S3G–S3I Fig). We conclude that Ama controls the number of leg myoblasts in an Nrt-independent way during larval stages.

## Ama is a potential downstream effector of FGF pathway in the regulation of myoblast number

A previous study proposed that Ama could modulate the RTK pathway activity during *Drosophila* brain development by regulating Sprouty (Sty) expression, a negative regulator of this pathway [22]. In this study, when Ama was specifically attenuated in glial cells, single-cell RNA-sequencing showed a strong down-regulation of cell cycle genes such as *PCNA* and *Cyclins*, as well as high expression of the pro-apoptotic gene *hid*, supporting a role of Ama in regulating cell survival and proliferation. Interestingly, we had previously shown that the RTK

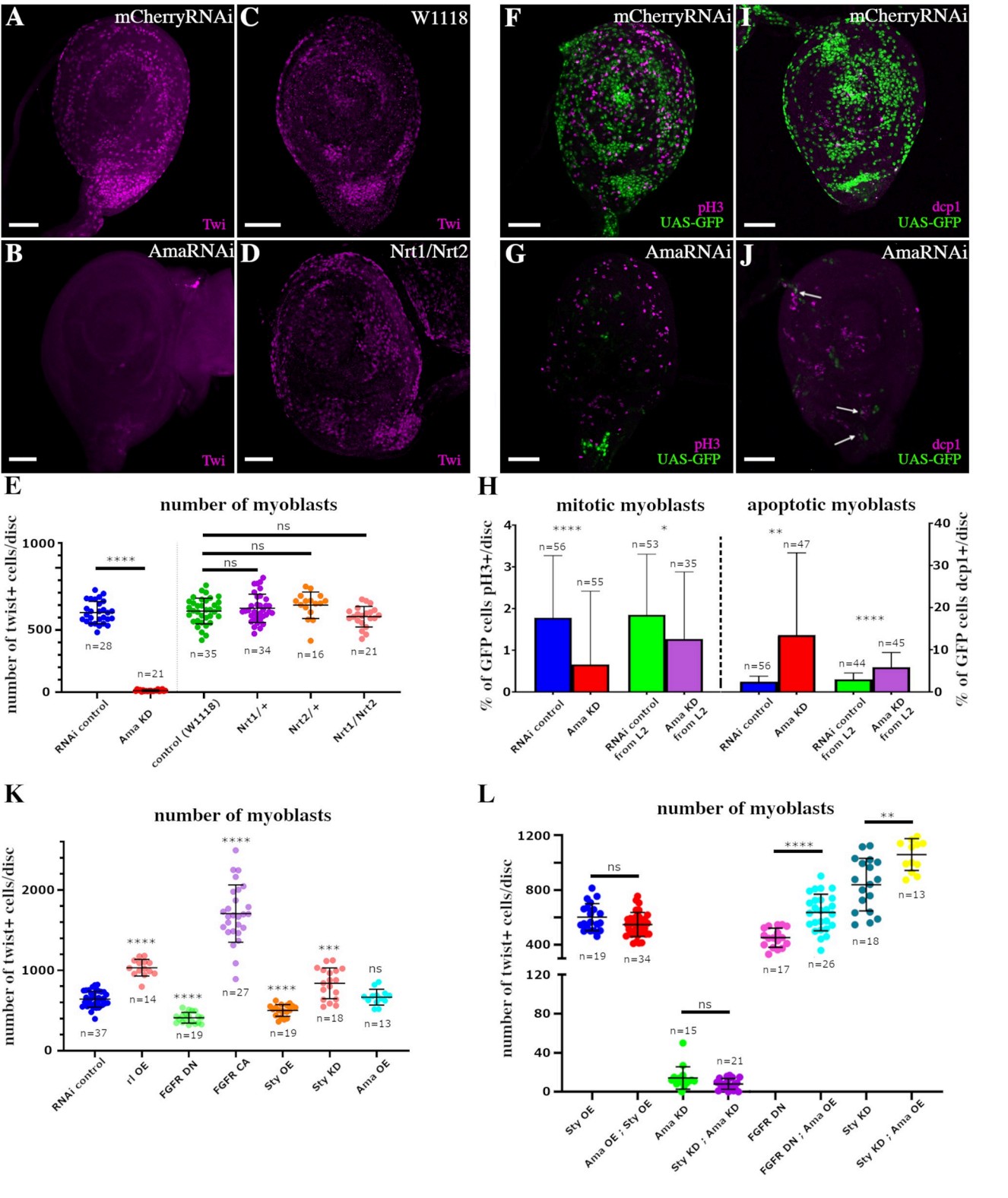

**Fig 4. Ama controls the pool of myoblast number independently of Nrt. (A–D)** L3 leg discs immunostained for Twi (magenta). **(A, B)** Expression of
*UAS-AmaRNAi* in myoblasts using R32D05-Gal4 driver leads to strong depletion of myoblasts when compared to R32D05-Gal4>UAS-mCherryRNAi control.

(C, D) Comparison between leg discs of w[1118] control and Nrt[1]/Nrt[2] transheterozygous show no difference in myoblast number. (E) Dot-plot graph showing the mean number of Twi-positive myoblasts per disc, in R32D05-Gal4>UAS-mCherryRNAi (control) leg disc versus R32D05-Gal4>UAS-AmaRNAi (Ama KD) leg disc, and between w[1118]control leg discs *versus* Nrt[1]/Nrt[+] and Nrt[2]/Nrt[+] heterozygous leg discs and Nrt[1]/Nrt[2] transheterozygous leg discs. (F, G) R32D05-Gal4>UAS-GFP (green) leg discs immunostained for pH3 (magenta); a small portion of myoblasts are proliferating in control (F), while only few cells remained after *UAS-AmaRNAi* expression in the myoblasts (G); none of them are pH3-positive. (I, J) R32D05-Gal4>UAS-GFP (green) leg discs immunostained for dcp1 (magenta); several apoptotic cells can be found among the GFP-positive cells remaining after *UAS-AmaRNAi* expression (arrows in J). (H) Graphs showing the percentage of mitotic myoblasts (on left) and the percentage of apoptotic myoblasts (on right). Compared to UAS-mCherryRNAi (control), the percentage of mitotic myoblasts is significantly reduced when *UAS-AmaRNAi* is expressed in the myoblasts from early larval stages (Ama KD) and when it is expressed from the beginning of L2 stage using Gal80[ts] (Ama KD from L2). Compared to UAS-mCherryRNAi (control), the percentage of apoptotic myoblasts is significantly higher when *UAS-AmaRNAi* is expressed in the myoblasts from early larval stages (Ama KD) and when it is expressed from the beginning of L2 stage using Gal80[ts] (Ama KD from L2). (K) Dot-plot graph showing the mean number of Twi-positive myoblasts per disc, in R32D05-Gal4 leg discs crossed with different UAS-transgenic lines affecting the FGFR pathway. Statistical analysis reveals an increase in the total of myoblasts when overexpressing an activated ERK (rl OE), a constitutive active FGFR (FGFR OE) or when down-regulating *sty* expression (Sty KD), compared to control RNAi. Inversely, overexpressing *sty* (Sty OE) or a dominant negative FGFR (FGFR DN) reduce the number of myoblasts. Note that overexpressing *Ama* (Ama OE) is not sufficient to induce an increase of myoblast number. (L) Dot-plot graph showing the mean number of Twi-positive myoblasts per disc in rescue experiments using R32D05-Gal4 driver. Overexpressing *UAS-Sty* together with *UAS-Sty* (Ama OE; Sty OE) show no significant difference in myoblast number when compared to *UAS-LacZ, UAS-Sty* overexpression (Sty OE). Co-expressing *UAS-StyRNAi* with *UAS-AmaRNAi* (Sty KD; Ama KD) is not sufficient to rescue the loss of myoblasts after *UAS-AmaRNAi*; UAS-LacZ (Ama KD) expression. Co-expressing UAS-Ama with UAS-HtlDN (FGFR DN; Ama OE) partially rescues the decrease of myoblast number of *UAS-HtlDN;* UAS-mCherryRNAi (FGFR DN) expression. Co-expression of *UAS-Ama* and *UAS-StyRNAi* (Sty KD; Ama OE) together enhances the phenotype of UAS-StyRNAi (Sty KD). In all graphs, error bars represent SD; *$p < 0.05$, **$p < 0.01$, ***$p < 0.001$, and ****$p < 0.0001$, ns: not significant, (Mann–Whitney test). $n$ = number of discs (S2 Data). Scale bars = 50 μm.

pathway, FGFR, is involved in regulating the number of leg disc myoblasts [10]. To test whether Ama could regulate myoblast number through the modulation of the RTK pathway, we conducted a set of gain- and loss-of-function experiments using the R32D05-Gal4 line. First, we counted the total number of Twist+ myoblasts in leg disc expressing an active form of the downstream effector of the RTK pathway, ERK, also named Rolled (UAS-rl[sem]), a dominant-negative form (UAS-Htl[DN]) or a constitutively active form (UAS-Htl[CA]) of the FGF receptor Heartless (Htl) (Fig 4K and S2 Data). Compared to control (mean = 643 myoblasts/disc, $n = 37$), we found that the number of myoblasts was significantly increased in rl[sem] (mean = 1032, $n = 14$) and in Htl[CA] (mean = 1,706, $n = 27$) expressing leg discs, and reduced after Htl[DN] induction in myoblasts (mean = 410, $n = 19$). This result confirmed that RTK pathways, and notably the FGFR pathway, are involved in controlling the number of total myoblasts. Overexpression of Sty (UAS-Sty) resulted in a significant decrease of myoblast number (mean = 502, $n = 19$), whereas its reduction (UAS-StyRNAi) leads to more myoblasts (mean = 839, $n = 18$), in accordance with its role of inhibiting FGFR pathway [20].

As Ama negatively regulates Sty expression in glial cells, we asked whether Ama could counteract Sty-induced changes in myoblast numbers in the developing leg. We found that Ama overexpression (Ama-HA) alone cannot rescue the reduced number of myoblasts caused by Sty overexpression (UAS-lacZ; UAS-Sty) (Fig 4L and S2 Data). Similarly, no rescue was observed when co-expressing AmaRNAi with StyRNAi compared to UAS-lacZ; UAS-AmaRNAi (Fig 4L). These results indicate that the hypothesis that Ama stimulates the FGFR pathway by negatively regulating *Sty* expression is not sufficient to explain the role of Ama in controlling leg myoblast number (see Material and methods for myoblast counting details). Nevertheless, we continued to investigate possible connections between Ama and the FGFR pathway. Previous work has shown that the expression of the constitutively active form of ERK can rescue the phenotype of Ama depletion in glial cells [22]. Therefore, we tried to rescue the UAS-AmaRNAi phenotype by co-expressing the UAS-rl[sem]. However, rl[sem] overexpression was insufficient to rescue myoblast depletion (Fig 4L). Since we could not show an upstream effect of Ama on the FGFR pathway, we asked if Ama could act downstream. Strikingly, leg discs co-expressing UAS-AmaHA and UAS-Htl[DN] showed a significantly higher number of myoblasts compared to the ones expressing UAS-Htl[DN]; UAS-mcherryRNAi. This suggests that Ama could, in fact, be a target of the FGFR pathway. Lastly, when Ama was overexpressed

in a context where Sty expression was decreased (UAS-StyRNAi; UAS-AmaHA), the increase in the number of myoblasts was even higher than when only Sty expression was reduced (Fig 4L). However, the mere overexpression of Ama (UAS-AmaHA) without modifying Sty expression had no impact on the number of myoblasts (Fig 4K).

Thus, our results indicate that Ama does not promote myoblast proliferation by alleviating Sprouty's inhibitory effect on the RTK pathway, in contrast to the previously described effect on glial cells [22]. However, the fact that Ama overexpression and Sty knockdown show an additive effect on myoblasts number and that Ama overexpression could improve loss of myoblasts in the Htl^DN context supports a model in which Ama acts downstream of FGF/RTK pathway. Another, non-mutually exclusive, possibility is that Ama enhances myoblast responsiveness to proliferation-inducing FGF signals. This view is supported by 2 studies. First, Ama knockdown was found to reduce the suppressive effect on retinal degeneration of the E3 ligase SORDD1 [35]. Second, a two-hybrid screen found Ama interaction with Syndecan [36], a proteoglycan acting as a co-receptor that modulates ligand availability for the FGF receptor [37].

## Ama and its receptor Nrt are required for myoblast-tendon adhesion

Given the restricted *Nrt* expression in one unique tendon precursor, the tilt, and the known role of Nrt as an adhesion molecule [13–15], we hypothesized that Ama and Nrt interact to ensure adhesion between myoblasts and the tilt. To test this hypothesis, we used an alternative myoblast specific driver, R15B03-Gal4 (S1A–S1F Fig), to induce Ama attenuation (UAS-AmaRNAi) after the extensive larval phases of proliferation. We combined the R15B03-Gal4 driver with R79D08-lexA that induced lexAop-GFP transgene in tendon cells (S4G–S4I Fig), allowing us to observe the developing tendons while inducing UAS-AmaRNAi in myoblasts from late L3 stage. We then analyzed the consequences of late myoblast-specific *Ama* knockdown on myoblast distribution at 5 h APF, a time when myoblasts are properly aligned along the elongating tilt tendon (Soler and colleagues, Fig 5A and 5A'). Late myoblast-specific *Ama* knockdown results in an apparent misdistribution of the myoblasts around the tilt tendon (Fig 5B and 5B' compared to Fig 5A and 5A'). To quantify the myoblast position variability relative to the tilt, we measured the shortest distance between each myoblast and the surface of the tilt and calculated the myoblast-to-tendon mean distance (dMT) for each disc (for detailed protocol, see Material and methods and S5 Fig). In controls, myoblasts are located at an overall average dMT of 4 μm from the tilt across all discs, whereas in AmaRNAi-expressing leg discs, this value increased slightly but significantly to 5.4 μm. This result indicates that *Ama* knockdown, leads to myoblasts mis-distribution around the tilt, suggesting that the reduction of Ama level could affect myoblast-tendon adhesion (Fig 5D and S4 Data). To better assess whether *Ama* knockdown increases the probability of the myoblasts being located farther away from the tilt than in control, we calculated the percentage of discs for which the dMT was greater than 4 μm (average dMT across all control discs), and 68% of the discs expressing R15B03-Gal4>AmaRNAi had a dMT greater than 4 μm, compared to 45% of the control discs (Fig 5E and S4 Data). Thus, reducing level of *Ama* expressed by the myoblasts significantly increases the proportion of discs in which myoblasts are not properly associated with the tilt. In parallel, we generated a strain carrying the *Nrt^2* allele and the R79D08-lexA>lexAop-GFP transgenes that we crossed with the line carrying the *Nrt^1* allele. F1 were then analyzed the same way as Ama knockdowns for myoblast positioning relative to the tilt. R79D08-lexA>lexAop-GFP; *Nrt^2*/*Nrt^1* transheterozygous mutants exhibit an average dMT of 6.2 μm, significantly elevated compared to the average dMT of R79D08-lexA>lexAop-GFP of 4.3 μm and R79D08-lexA>lexAop-GFP; *Nrt^2*/*Nrt^+* controls (4.8 μm) (Fig 5D). As expected, the percentage of discs with a dMT greater than the average dMT of control discs (4.3 μm) was much higher

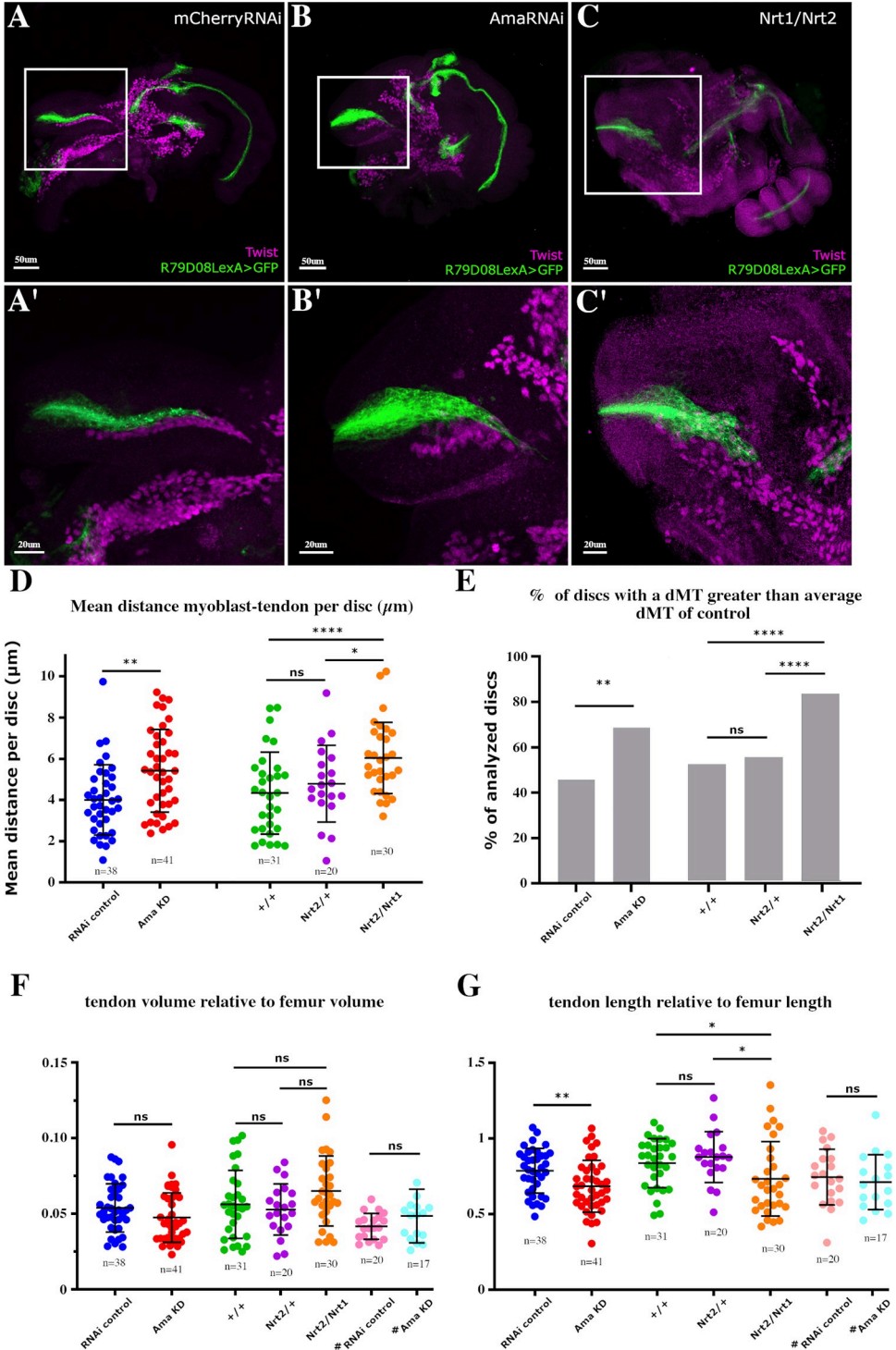

**Fig 5. Myoblast positioning and tendon morphogenesis require Ama and Nrt. (A–C)** Confocal optical sections of 5 h APF leg disc immunostained for Twi from **(A)** R79D08-lexA>lexAop-mCD8::GFP; R15B04-Gal4>UAS-mCherryRNAi, from **(B)** R79D08-lexA>lexAop-mCD8::GFP; R15B04-Gal4>UAS-AmaRNAi and from **(C)** R79D08-lexA>lexAop-mCD8::GFP; Nrt$^1$/Nrt$^2$ transheterozygous. **(A'–C')** Higher magnifications of the dorsal femur regions from (A), (B), and (C), respectively. **(A, A')** In control leg discs expressing *UAS-mCherryRNAi* using R15B03-Gal4 late myoblast driver, the tilt (green) has elongated within the dorsal femur cavity to form a long internal structure along which the myoblasts (magenta) are aligned. **(B, B')** When *UAS-AmaRNAi* is expressed in the myoblasts, they lose their adhesion with the tilt; the tilt itself appears wider and shorter compared to control. **(C, C')**

The same observations can be made in Nrt[1]/Nrt[2] transheterozygous leg discs, with a misdistribution of the myoblasts along the tilt and an elongation default of the tilt. **(D)** Dot-plot graph showing the mean distance between myoblasts and the tilt surface; each dot corresponds to the mean distance between the tilt and all the myoblasts for one disc. The average of mean distance for R79D08-lexA>lexAop-mCD8::GFP; R15B04-Gal4>UAS-AmaRNAi (Ama KD) leg discs is statistically higher compared to R79D08-lexA>lexAop-mCD8::GFP; R15B04-Gal4>UAS-mCherryRNAi (control RNAi) leg disc ($p < 0.0016$). This average is also higher when comparing R79D08-lexA>lexAop-mCD8::GFP; Nrt[1]/Nrt[2] transheterozygous leg discs with both R79D08-lexA>lexAop-mCD8::GFP (+/+) ($p < 0.0008$) and R79D08-lexA>lexAop-mCD8::GFP; Nrt[2]/Nrt[+] ($p < 0.0201$) control leg discs. **(E)** Graphs showing the percentage of discs for which the mean distance between the myoblasts and the tilt is higher than the average mean distance of the controls UAS-mCherryRNAi and +/+, respectively. **(F)** Dot-plot graph showing the volume of the tilt, each dot corresponds to one disc. No statistically significant difference is observed between the control lines and the lines expressing *AmaRNAi*, as well as the mutant lines for Nrt. **(G)** Dot-plot graph showing the lengths of the tilt, each dot corresponds to one disc. *UAS-AmaRNAi* (Ama KD) myoblast late expression leads to shortening of the tilt length compared to UAS-mCherryRNAi control ($p < 0,0042$). This shortening is also evident in transheterozygous mutants Nrt[1]/Nrt[2] compared to wild-type Nrt homozygous ($p < 0,025$) and Nrt[2]/ Nrt[+] ($p < 0,011$). #Tendon-specific AmaKD in R79D08-lexA>lexAop-mCD8::GFP; Sr-Gal4>UAS-AmaRNAi discs does not significantly affect either the volume or length of tendons. In all graphs, error bars represent SD; *$p < 0.05$, **$p < 0.01$, ***$p < 0.001$, and ****$p < 0.0001$, ns: not significant, Mann–Whitney test (D) and (G), Fisher's exact test (E). $n$ = number of discs; (S4 Data). APF, after pupae formation.

in transheterozygous mutants (84%) compared to controls (48% and 45%, respectively, Fig 5E). Thus, in both *Ama* and *Nrt* knockdown contexts, myoblasts lose their ability to aggregate and to align along their corresponding tendon. These results support the hypothesis that Ama/Nrt heterophilic interactions play a role in myoblasts for tendon precursor adhesion. Given that Ama is a secreted protein [18] and Nrt is only present on the surface of the tendon, it is likely that Ama interacts with another membrane-associated protein at the surface of the myoblasts. Alternatively, the presence of a short hydrophobic domain at its COOH-terminal end suggests that Ama could directly tether to the membrane via a glycosyl-phosphotidylinositol (GPI) anchor [18] similar to other cell adhesion molecules [38].

Thus, Ama could mediate tendon-myoblast adhesion by interacting with Nrt on the tendon side and myoblast-myoblast adhesion on the other side, directly or through another unidentified transmembrane receptor. Ama is orthologous to several Ig-like neuronal cell adhesion molecules in vertebrates, including the IgLON 4 and 5 GPI-anchored membrane proteins that regulate myoblast adhesion during myogenesis and muscle regeneration by providing an essential microenvironment that ensures muscle stem cell survival [39,40].

Thus, the conservation of Ama's function could highlight a pivotal role in sustaining muscle tissue homeostasis and development across evolution.

## Ama and Nrt contribute to long tendon morphogenesis

In addition to the loss of myoblast adhesion observed in Nrt mutants and R15B03-Gal4>AmaRNAi discs, we also noticed a general alteration in the morphology of the tilt. In particular, the tilt appears shorter in mutants compared to controls (Fig 5B' and 5C'). Suggesting that in addition to their role in myoblast positioning, Ama-Nrt interaction would also contribute to tendon morphogenesis. This would support the hypothesis of a reciprocal influence between these 2 cell types for coordinating their developmental fate.

Given the implications of Ama and Nrt in glial cell migration and axonal pathfinding [16,22,41], and because leg long tendons form by collective migration of epithelial cells [25], we wondered if loss of Ama or Nrt could affect tilt elongation. We measured the length of the tilt relative to the length of the femur (see Material and methods), and found that tilts in discs with *Ama* knockdown in myoblasts are shorter than in control (Fig 5G and S4 Data). In a similar manner, the combination of transheterozygous Nrt[1]/Nrt[2] mutants exhibit a shorter tilt compared to controls (Fig 5G). Interestingly, in both cases, the overall tendon volume is not

affected. This result implies that the tendon shortening is somehow compensated for by a widening of the structure, resulting in a clearly visible deformation of the tendon (Fig 5F and S4 Data). As we showed that tendon cells also express *Ama*, we investigated the effect of tendon-specific *Ama* knockdown on tendon elongation. Tendon-specific induction of UAS-AmaR-NAi using Sr-Gal4 driver did not result in any tendon elongation defects (Fig 5F and 5G). Thus, Ama secretion by the myoblasts, likely interacting with Nrt at the tendon surface, is essential for proper elongation and morphogenesis of the tilt long tendon.

Altogether, these results support a model in which Ama::Nrt interaction is important for ensuring specificity of myoblast to tendon adhesion. These results also suggest that, in return, Ama could provide a guiding cue through Nrt that would be necessary for the correct growth of the tendon. It has been shown that Ama and Nrt are dominant enhancers of the axon misguidance phenotype observed in an Abelson tyrosine kinase (Abl) mutant [16]. Because Abl regulates modulators of the actin cytoskeleton [42–44] is tempting to speculate that activation of this pathway by Ama/Nrt could contribute to tendon growth in coordination with myoblast adhesion.

## Conclusion

Overall, we identify *Ama* and *Nrt* as a part of transcriptional signature of developing leg muscle and tendon precursors. We show that Ama is a key player with a double role during appendicular myogenesis. First, during early larval development, Ama acts in a myoblast-autonomous way to promote their proliferation and to establish the appropriate number of leg myoblasts, independentlty of Nrt. Our genetic analyses indicate that Ama acts downstream of FGFR pathway to fulfill this function. Second, slightly later in development, Ama secreted by myoblasts interacts with Nrt expressed at the surface of one of leg tendon (tilt) precursors. This interaction is crucial for myoblast-tendon specific adhesion, tilt tendon elongation, and consequently, coordinated development of tilt-associated appendicular muscle. Thus, Ama and Nrt represent a pair of molecules that ensure early cross talk between 2 key components of developing muscle, the tendon precursors and myoblasts. This result points to the importance of reciprocal cell–cell communication during early steps of muscle development. We also uncover a dual role for Ama in regulating both myoblast number and cell adhesion, indicating that within a cell population the use of a single molecule can be reoriented by modulating its mode of interaction. Giving the well-characterized cellular and molecular convergences observed during muscle development between vertebrates and *Drosophila*, our findings could be relevant to analogous processes in vertebrates.

## Material and methods

### Fly strains and genetic

The following fly stocks were used: enhancer traps line sr-gal4$^{md710}$ ([45], BDSC 2663); UAS-Lifeact.GFP (BDSC 35544); UAS-mCherryCAAX (BDSC 59021); R32D05-Gal4 (BDSC 49712); UAS-stingerGFP (BDSC 84278); UAS-mCherryRNAi (BDSC 35785); UAS-AmaRNAi (BDSC 33416); w$^{1118}$; Nrt$^1$/TM6B (BDSC 25032); Nrt$^2$/TM6B (BDSC 25033); tubGal80$^{ts}$ (BDSC 7108); UAS-rlSem (BDSC 59006); UAS-Htl-DN (BDSC 5366); UAS-Htl-CA (BDSC 5367); UAS-StyRNAi (VDRC 330208); UAS-Sty (BDSC 20670); UAS-lacZ (BDSC 1776); R15B03-Gal4 (BDSC 49261); R79D08-lexA (BDSC 54734); lexAop-mCD8::GFP (BDSC 66545); UAS-mCherryNLS (BDSC 38425). For experiments using tubGal80$^{ts}$, crosses were kept at 18°C and then shifted at the appropriate stage (L2) to 30°C. All other experiments were performed at 25°C.

## Cell sorting and RNA extraction

The fluorescence-activated cell sorting and RNA sequencing of 0 h APF leg disc myoblasts were conducted following the same protocol as described in Laurichesse and colleagues. For myoblast sorting, we used the targeted expression system lexA/lexAop to express the lex-AopGFP transgene specifically in leg disc myoblasts (R32D05-lexA>lexAopGFP).

## RNA-seq and transcriptomic data analysis

Extracted total RNA from GFP+ myoblasts and input (IP) cells (sample of all leg disc cells) from 3 different replicates were sent to Heidelberg genomic platform EMBL for high-through-put mRNA sequencing (NextSeq 500/Illumina). They generated mRNA libraries with NEB RNA Ultra kit (New England Biolabs E7770L) and oligo(dT) probe were used to target mRNA for cDNA synthesis. Single-end multiplexing was performed. Bioinformatic analysis was performed using FastQC and Cutadapt [46] for reads quality control and filtration. Reads alignment and counting were performed using Hisat2 [47] and HTseq [48] (reference genome Dm6 Ensembl release 70). Values were normalized for each gene (RPKM) using Cufflinks [49] and differential expression between IP and GFP+ samples were determined using R script DEseq2 V: 3.13 software. Fold change (FC) between GFP+ and IP samples was computed using their normalized raw counts. GEO accession number: GSE245192.

## Ama transgenes

**UAS-Ama-HA transgene.** The UAS-Ama C-terminal tagged with FLAG-HA (UFO01101 —DGRC Stock 1621050; https://dgrc.bio.indiana.edu//stock/1621050; RRID:DGRC_1621050) clone was obtained from the Universal Proteomics Resources, a Berkeley Drosophila Genome Project (BDGP) (https://www.fruitfly.org/EST/proteomics.shtml) and contains a *white* gene selectable marker and an *attB* site. The PhiC31 transformation services in BestGene (http://www.thebestgene.com/) were performed on *y¹ w⁶⁷ᶜ²³; P{CaryP}attP2* line (BDSC 8622) with an estimated CytoSite 68A4 (chromosome 3) and on *y¹ w⁶⁷ᶜ²³; P{CaryP}attP40* line with an estimated CytoSite 25C6 (chromosome 2) using the UAS-Ama construct above.

**Ama::EGFP transgene.** w¹¹¹⁸ line expressing EGFP C-terminal tagged endogenous Ama protein was generated by CRISPR-mediated-genome editing by homology repair using the following guide 5′- CTTCGAGGGCACTGGGATGACGGTC-3′ by Wellgenetics (Taiwan). Detailed protocol on request.

## Immuno-histochemistry and confocal microscopy

The following primary antibodies were used: mouse anti-Nrt (1:100—BP106- Developmental Studies Hybridoma Bank DSHB), rabbit anti-Twist (1:300—our lab), rabbit anti-Dcp1 (1:100 —Cell Signaling), rabbit anti-pH3 (1:500—Merk Millipore), goat anti-GFP (1:500 –ab5450-Abcam), and rabbit anti-Ama (1:500, gift from I. Silman). Secondary antibodies donkey anti-rabbit, donkey anti-mouse conjugated to cy3- or cy5- fluorochromes (Jackson Immunoresearch), donkey anti-rabbit Dylight 405 (Jackson Immunoresearch), donkey anti-rabbit Alexa Fluor 488 (Thermo Fisher), and donkey anti-goat Alexa Fluor 488 (Invitrogen) were used (1:500). Immunohistochemistry experiments were performed on samples fixed in 4% formaldehyde (FA-methanol free) for 20 min, washed in 0.5% PBS-Triton, and blocked in 30% horse serum before immunostaining with primary antibodies, at 4°C overnight. The samples were washed in 0.5% PBS-Triton and incubated with secondary antibodies, at room temperature during 2 h. After 3 washes in 1XPBS, samples are mounted in Vectashield medium for analysis.

Immunostained samples were imaged on LSM800 Zeiss confocal microscope, and images were analyzed using Imaris 9.8.2 and ImageJ-fiji 2.9.0/1.53t softwares.

## In situ hybridization

Leg imaginal discs were dissected in 1XPBS and fixed by 4% formaldehyde (FA-methanol free) for 20 min. Samples were washed 3 times in 1XPBS and incubated in methanol over-night (o/n) at −20˚C. Samples were progressively rehydrated in 75% MetOH-25% PBS for 5 min, 50% MetOH-50% PBS for 5 min, 25% MetOH-75% PBS for 5 min, 10% MetOH-90% PBS for 5 min, and samples were quenched with 0.3% $H_2O_2$ for 15 min. The samples were washed 3 times in 1XPBS for 5 min each and were permeabilized by 80% acetone for 10 min at −20˚C, and washed 3 times in PBS-Tween (PBSTw) for 5 min at room temperature. The samples were refixed by 4% FA for 20 min and washed 3 times for 5 min each. Then, samples were incubated in 50% Hybridization Buffer (50% formamide, 2× SSC, 1 mg/ml Torula RNA, 0.05 mg/ml Heparin, 2% Roche blocking reagent, 0.1% CHAPS, 5 mM EDTA, 0.1% Tween 20)– 50% PBSTw for 5 min, 75% HB-25% PBSTw for 5 min, 90% HB-10% PBSTw for 5 min, and incubated during 1.5 h at 60˚C in 100% HB Buffer. The samples were incubated o/n at 60˚C with anti-sense DIG-labeled RNA probes against *Ama* diluted in HB (1:100). The samples were washed in 100% HB for 1 h at 60˚C, in 75% HB-25% PBSTw for 15 min at 60˚C, in 50% HB-50% PBSTw for 15 min at 60˚C, in 25% HB-75% PBSTw for 15 min at 60˚C, in 10% HB-90% PBSTw at 60˚C for 15 min and were washed 3 times in PBSTw 5 min for each at room temperature. Then, the samples were incubated for 1.5 h in PBSTw-1%BSA before being incubated with anti-DIG-POD antibody (Roche, 1:2,000) and other primary antibodies o/n at 4˚C. After 5 washes in PBSTw, Tyramide Signal Amplification (TSA) were performed in 100 μl Amplification Buffer + 2 μl TSA$^+$ Cy3 (PerkinElmer, 1:50) for 5 min at room temperature. The sample were washed 6 times for 5 min each in PBSTw and were incubated with secondary antibodies for 2 h at room temperature. After 3 washes in 1XPBS, samples are mounted in Vectashield medium for analysis.

## Quantification and statistical analysis

**Myoblast counting.** Images were processed with Imaris 9.8.2 software and GFP+, Twist+, Dcp1+, and pH3$^+$ cells were detected automatically using the "*Spots*" function. The automatic cell counting was subject to manual adjustment. Myoblast proliferation and apoptosis were calculated by determining the percentage of pH3+/GFP+ cells and Dcp1+/GFP+ cells, respectively. It is noteworthy that the use of 2 UAS-transgenes (UAS-lacZ; UAS-Sty) is sufficient to rescue the reduction of myoblast number induced by UAS-Sty expression alone, emphasizing the importance of adding a second UAS-transgene as a control.

**Myoblasts to tendon distance.** Tilt surface was automatically determined using the "*Surface*" function of Imaris Software, then a distance map was generated using the "*Distance Transformation*" function. The distance in μm between the tilt surface and each myoblast, detected using the "*Spot*" function as described above, was calculated with the "*intensity center spot*" function (see also S5 Fig).

**Tendon length and volume measurement.** Contour surfaces of the tilt and of the femur were determined either automatically (for the tilt) or manually (for femur) using Imaris "*Surface*" function to extract 3D objects and calculate their volumes. Tilt volume was determined by normalizing the volume of the tilt over the volume of the femur. 3D objects were then used to measure tilt and femur lengths using "*skeletonize*" *(2D-3D)* plugin of ImageJ-Fiji 2.9.0/1.53t software. Tilt length was then determined by normalizing the length of the tilt over the length of the femur.

Statistical analyses were carried out using Prism 9.2.0 and statistical significances were determined using Mann–Whitney tests or Fisher's exact test/Chi-square test. Differences were considered significant if $p$-value $<0.05$.

## Supporting information

**S1 Fig.** *Nrt* and *Ama* **expression patterns in L2 leg disc and** *R32D05-Gal4* **expression pattern in L3 leg discs. (A–C)** Confocal optical sections of Ama::EGFP (gray) L2 leg discs immunostained for Twist (red). Ama::EGFP protein is found surrounding most of the myoblasts. **(D–F)** Confocal optical sections of Sr-Gal4>UAS-Lifeact.GFP (green) L2 leg discs immunostained for Nrt (magenta). Tendon precursors are not yet specified at this stage and no Nrt presence can be detected. **(G–I)** R32D05-Gal4 drives expression in all leg disc myoblasts. *Gal4* expression under the control of DNA sequences from regulatory regions of *string* gene drives the expression of *GFP* (green) that colocalizes with Twi-positive myoblasts (magenta). This expression was already observed in early L2 stage (S3 Fig).
(TIF)

**S2 Fig. During early metamorphosis** *Ama* **is expressed in both myoblasts and the tilt tendon. (A–F)** In situ hybridizations of *Ama* (gray) in dorsal femur region from R32D05-Gal4>UAS-GFP (green) 5 h APF leg discs. **(A–C)** In control disc expressing *UAS-mCherryRNAi* in myoblasts, *Ama* expression is detected in both myoblasts (in green) and the elongating tilt tendon (red outlined). **(D–F)** When AmaRNAi is specifically driven in the myoblasts using R32D05-Gal4, *Ama* expression remains visible only in the elongating tendon, indicating that, at this stage, both myoblasts and the tilt express *Ama*. **(G–L)** In situ hybridizations of *Ama* (gray) in dorsal femur region from Sr-Gal4>UAS-LifeactGFP (green) 5 h APF leg discs. **(G–I)** In control disc expressing *UAS-mCherryRNAi* in tendon cells, *Ama* expression is detected in both the elongating tilt tendon (green, red outlined) and the surrounding myoblasts. **(J–L)** When AmaRNAi is specifically driven in the tendon cells using Sr-Gal4, *Ama* expression is lost in the tilt but remains visible in the surrounding myoblasts, confirming that, at this stage, both myoblasts and the tilt express *Ama*. **(M–P)** Confocal sections of 5 h APF leg discs immunostained for Ama protein (gray) and FasIII (magenta) that allows to reveal the leg disc epithelium and the invaginating tilt (arrowheads). **(M, N)** In R32D05-Gal4>UAS-mcherryRNAi control leg discs, Ama protein is widely distributed in the dorsal femur region. **(O, P)** In R32D05-Gal4 leg discs expressing *UAS-AmaRNAi* specifically in the myoblasts, Ama secreted protein is barely visible (number of analyzed discs, AmaRNAi = 10, mCherryRNAi = 10).
(TIF)

**S3 Fig. Ama specific down-regulation in tendon and myoblasts. (A–C)** Tendon-specific *Ama* down-regulation does not affect proliferation rate and viability of tendon cells. Representative confocal images of the tilt in the dorsal femur of Sr-Gal4>UAS-mCherryNLS leg discs expressing UAS-AmaRNAi, immunostained for pH3 **(A)**, and Dcp1 **(B)** at L3 and 5 h APF stage, respectively. **(C)** As previously described, only rare mitotic events can be observed among leg tendon cells which are recruited from the leg disc epithelium (Laurichesse and colleagues). Consequently, no proliferative rate differences are observed between AmaKD and control leg discs. Similarly, we did not observe any significant increase of cell death in AmaKD in L3 or 5 h APF leg discs (S3 Data). **(D–L) Ama down-regulation leads to myoblast depletion in second larval instar before any** *Nrt* **expression is observed.** Confocal optical sections of R32D05-Gal4>UAS-GFP (green) leg discs at mid L2 stage immunostained for Twi **(D, G)** and for Nrt **(J)**. **(D–F)** In control leg discs, R32D05-Gal4 induces *GFP* expression (Green) in

all Twi-positive myoblasts (magenta). **(G, H)** *AmaRNAi* expression using R32D05-Gal4 driver leads to the loss of most of the myoblasts. **(J–L)** Nrt protein (magenta) is not detected in midL2 stage.
(TIF)

**S4 Fig. Expression patterns of R15B03-Gal4 and R79D08-lexA drivers. (A–F)** R15B03-Gal4>UAS-mCherryNLS (green) leg disc immunostained for Twi (magenta). **(A–C)** L2 leg disc shows no *mCherryNLS* expression in Twi-positive myoblasts, indicating that R15B03-Gal4 driver is not expressed in myoblasts at this stage. **(D–F)** During L3 stage, *mCherryNLS* starts to be expressed in Twi-positive myoblasts as well as in few other cells. **(G, H)** 5 h APF R79D08-lexA>lexAop-mCD8::GFP; Sr-Gal4>UAS-CAAXmCherry leg discs, R79D08—lexA driving lexAop-GFP allows the visualization of developing tendons identified through the expression of the well characterized *Sr-Gal4* driver expression (magenta).
(TIF)

**S5 Fig. Experimental determination of myoblasts to tendon distance using Imaris Software. (A)** Confocal optical sections of 5 h APF leg disc corresponding to the image presented in Fig 5 A and A′. R79D08-lexA>lexAop-mCD8::GFP allows the visualization of the tilt (green), myoblasts are visualized by Twi immunostaining (magenta) and the whole femur epithelium by Fasciclin 3 immunostaining (cyan). **(B)** Modelization of the tilt and associated myoblasts using Imaris Software. The tilt surface (gray) was determined automatically from the green channel using the "Surface" function of Imaris Software. Each myoblast associated with tilt were detected automatically (red) from the magenta channel (Twist) using the "Spot" function. As shown in Fig 2O–P′, myoblast membranes are very closed to each other. Therefore, we found it more accurate to use the nuclei of myoblasts (stained with Twi antibody) than the cytoplasmic membrane to individually identify myoblasts. Moreover, we considered that the distances between the myoblast nuclei and the tendon surface accurately reflect the spatial relationship between these 2 cell types, given that the myoblast nucleus occupies most of the cell volume. **(C)** From the 3D modelization of the "tilt surface" object, a distance map (blue) was generated from the outside "tilt surface" object using the "Distance Transformation" function of Imaris Software, after converting our data from 8-bit to 32-bit. Then, the "tilt surface" object serves as the focal point of the distance map, enabling the calculation of distances from any external object (here, the red spots representing individual myoblasts) to the nearest boundaries of the surface object. **(D)** Higher magnification, depicting the distance (green line) between the surface of the tilt and each myoblast (center of each spot). The distance in μm between the tilt surface and each myoblast is shown in spots statistics "intensity center channel" (channel corresponding to the distance map). This corresponds to the distance between the center of the spot (myoblast nucleus) and the surface of the "tilt surface" object closest to the spot (green line). The mean of all distances was then calculated.
(TIF)

**S1 Data. Differential gene expression analysis table.**
(XLSX)

**S2 Data. Individual quantitative obervations of myoblast number, proliferation, and apoptosis.**
(XLSX)

**S3 Data. Individual quantitative obervations of tendon cell number, proliferation, and apoptosis.**
(XLSX)

**S4 Data. Individual quantitative obervations of tendon volume and length and myoblasts-tendon distances.**
(XLSX)

## Acknowledgments

We thank Bloomington *Drosophila* Stock Center (BDSC) and Vienna *Drosophila* Resource Center (VDRC) for keeping and sending *Drosophila* stocks. We thank the Drosophila Genomics Resource Center (NIH Grant 2P40OD010949) for DNA clone. We thank the Bioimaging platform (CLIC microscopy facility), the Single-Cell platform (S3C), and the Bio-Informatic (BIM) at the iGRed institute. We are grateful to I. Silman for its gift of Ama antibody.

## Author Contributions

**Conceptualization:** Cedric Soler.

**Data curation:** Yoan Renaud.

**Formal analysis:** Blandine Moucaud, Yoan Renaud, Cedric Soler.

**Funding acquisition:** Guillaume Junion, Cedric Soler.

**Investigation:** Blandine Moucaud, Elodie Prince, Cedric Soler.

**Methodology:** Blandine Moucaud, Elodie Prince, Elia Ragot, Yoan Renaud, Guillaume Junion, Cedric Soler.

**Resources:** Krzysztof Jagla.

**Supervision:** Krzysztof Jagla, Cedric Soler.

**Validation:** Cedric Soler.

**Visualization:** Blandine Moucaud, Elodie Prince, Elia Ragot, Cedric Soler.

**Writing – original draft:** Blandine Moucaud, Cedric Soler.

**Writing – review & editing:** Blandine Moucaud, Krzysztof Jagla, Guillaume Junion, Cedric Soler.

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
