## [Editor Report · Decision Letter 0]

28 Jun 2024

Dear Dr Soler, 

Thank you for submitting your revised manuscript via Review Commons entitled "Amalgam plays a dual role in controlling the number of leg muscle progenitors and regulating their interactions with developing tendon" for consideration as a Research Article by PLOS Biology.

Your manuscript has now been evaluated by the PLOS Biology editorial staff as well as by an academic editor with relevant expertise and I am writing to let you know that we would like to send your revision back to the reviewers.

However, before we can do that, we need you to complete your submission by providing the metadata that is required for full assessment. To this end, please login to Editorial Manager where you will find the paper in the 'Submissions Needing Revisions' folder on your homepage. Please click 'Revise Submission' from the Action Links and complete all additional questions in the submission questionnaire.

Once your full submission is complete, your paper will undergo a series of checks in preparation for peer review. After your manuscript has passed the checks it will be sent out for review. To provide the metadata for your submission, please Login to Editorial Manager (https://www.editorialmanager.com/pbiology) within two working days, i.e. by Jul 02 2024 11:59PM.

Kind regards,

Ines

--

Ines Alvarez-Garcia, PhD

Senior Editor

PLOS Biology

---

## [Decision Letter · Decision Letter 1]

2 Aug 2024

Dear Dr Soler,

Thank you for your patience while we considered via Review Commons your revised manuscript entitled "Amalgam plays a dual role in controlling the number of leg muscle progenitors and regulating their interactions with developing tendon" for publication as a Research Article at PLOS Biology. This revised version of your manuscript has been evaluated by the PLOS Biology editors, the Academic Editor and one of the original reviewers.

Based on the review, we are likely to accept this manuscript for publication, provided you address the data and other policy-related requests stated below.

In addition, we would like you to consider a suggestion to improve the title:

"Amalgam plays a dual role in controlling the number of leg muscle progenitors and regulating their interactions with the developing Drosophila tendon"

We expect to receive your revised manuscript within two weeks. 

*Published Peer Review History*

*Press*

Sincerely,

Ines

--

Ines Alvarez-Garcia, PhD

Senior Editor

PLOS Biology

Fig. 1A, B; Fig. 4E, H, K, L; Fig. 5D-G and Fig. S3C

CODE POLICY

DATA NOT SHOWN?

Reviewers' comments

Rev. 1:

I want to congratulate the authors for this interesting manuscript and their successful revisions.

I am happy to see that my suggestions were addressed accordingly and the new findings are nicely supporting the hypothesis put forward by the authors. This includes the reduced Ama staining at 5 h APF after muscle-specific knockdown of Ama, the improved high resolution imaging of the tendon precursors using a membrane label and importantly the fact that Ama in tendon does not contribute to the Ama mutant phenotype described here. So, the autonomous function of Ama in myoblasts was nicely backed-up by these experiments. The additional Suppl. Figure explaining the distance measurements between myoblast nuclei and tendon cells is also helpful.

In summary, my suggestions have been addressed and I recommend acceptance of this interesting paper.

---

## [Editor Report · Decision Letter 2]

14 Sep 2024

Dear Dr Soler,

Thank you for the submission of your revised Research Article entitled "Amalgam plays a dual role in controlling the number of leg muscle progenitors and regulating their interactions with the developing Drosophila tendon" for publication in PLOS Biology. On behalf of my colleagues and the Academic Editor, Simon Hughes, I am delighted to let you know that we can in principle accept your manuscript for publication, provided you address any remaining formatting and reporting issues. These will be detailed in an email you should receive within 2-3 business days from our colleagues in the journal operations team; no action is required from you until then. Please note that we will not be able to formally accept your manuscript and schedule it for publication until you have completed any requested changes.

PRESS

Sincerely, 

Ines

--

Ines Alvarez-Garcia, PhD

Senior Editor

PLOS Biology
